# New Insight into Plant Signaling: Extracellular ATP and Uncommon Nucleotides

**DOI:** 10.3390/cells9020345

**Published:** 2020-02-02

**Authors:** Małgorzata Pietrowska-Borek, Jędrzej Dobrogojski, Ewa Sobieszczuk-Nowicka, Sławomir Borek

**Affiliations:** 1Department of Biochemistry and Biotechnology, Faculty of Agronomy and Bioengineering, Poznań University of Life Sciences, Dojazd 11, 60-632 Poznań, Poland; dobrogojski@gmail.com; 2Department of Plant Physiology, Faculty of Biology, Adam Mickiewicz University, Poznań, Uniwersytetu Poznańskiego 6, 61-614 Poznań, Poland; evaanna@amu.edu.pl (E.S.-N.); borek@amu.edu.pl (S.B.)

**Keywords:** adenosine 5′-phosphoramidate, adenosine 5′-tetraphosphate, diadenosine 5′,5′′′-tetraphosphate, dinucleoside polyphosphates, eATP, eNAD(P)^+^

## Abstract

New players in plant signaling are described in detail in this review: extracellular ATP (eATP) and uncommon nucleotides such as dinucleoside polyphosphates (Np_n_N’s), adenosine 5′-phosphoramidate (NH_2_-pA), and extracellular NAD^+^ and NADP^+^ (eNAD(P)^+^). Recent molecular, physiological, and biochemical evidence implicating concurrently the signaling role of eATP, Np_n_N’s, and NH_2_-pA in plant biology and the mechanistic events in which they are involved are discussed. Numerous studies have shown that they are often universal signaling messengers, which trigger a signaling cascade in similar reactions and processes among different kingdoms. We also present here, not described elsewhere, a working model of the Np_n_N’ and NH_2_-pA signaling network in a plant cell where these nucleotides trigger induction of the phenylpropanoid and the isochorismic acid pathways yielding metabolites protecting the plant against various types of stresses. Through these signals, the plant responds to environmental stimuli by intensifying the production of various compounds, such as anthocyanins, lignin, stilbenes, and salicylic acid. Still, more research needs to be performed to identify signaling networks that involve uncommon nucleotides, followed by omic experiments to define network elements and processes that are controlled by these signals.

## 1. Introduction

Plant signaling is a set of phenomena that enables the transduction of external and internal signals into physiological responses such as modification of enzyme activity, cytoskeleton structure, and gene expression. It is known that in plants there exist mechanisms involved in the signal transduction pathways. Plants have evolved signaling networks providing reactions to environmental stimuli through signaling proteins such as plasma membrane receptors and ion transporters and by cascades of kinases and other enzymes up to effectors. For many years plant hormones were considered to be dominant molecules in plant signaling. Nowadays this term embraces many other compounds including second messengers, such as cytosolic Ca^2+^ [1], reactive oxygen (ROS) and nitrogen species (RNS) [2] or cyclic nucleotides such as adenosine 3′,5′-cyclic monophosphate (cAMP) and guanosine 3′,5′-cyclic monophosphate (cGMP) [3]. Nowadays there is more and more information about synthesis, degradation, and function of cAMP and cGMP in plant [3,4,5], and they are currently accepted as key signaling molecules in many processes in plants including growth and differentiation, photosynthesis, and biotic and abiotic defense [6]. However, recently it was shown that nucleotides, such as ATP and (di)nucleoside polyphosphates, also can play signaling roles in plant cells. In this review, we focus on the extracellular ATP (eATP) and uncommon nucleotides, such as mono- (p_n_Ns) and dinucleoside polyphosphates (Np_n_N’s), and their new function as signaling molecules.

## 2. eATP as a Signaling Molecule

Adenosine 5′-triphosphate (ATP), as well as the other nucleoside triphosphates, are established as agents providing energy in various reactions inside cells, both in animal and plant organisms [7]. As ATP is omnipresent in all living cells, it is often called the essential energy currency molecule. In the 1970s it was first hypothesized that ATP might be released into the extracellular environment and act as a signaling compound for animal cells [8]. However, it is worth mentioning that the first effect of eATP on a cell was noted much earlier, i.e., in 1929, during research on heart muscle contraction [9]. In animals, three possible ways of ATP release from the cells into the extracellular matrix have been proposed; they involve multiple channels, transporters, and exocytosis. The interest in the eATP signaling function accelerated after the first purinoreceptor was cloned and characterized in rat brain tissue [10]. Currently, eATP, as well as some of the other nucleotides, are considered as signaling molecules mediating numerous animal cellular processes. For decades the role of nucleotides as signaling molecule functioning similarly in plants as it was demonstrated in animals was viewed with skepticism. A real breakthrough in this topic came with the discovery of the existence of a plant transmembrane receptor protein with serine/threonine kinase activity having a high affinity for extracellular nucleotides [11]. However, the evidence for the mechanism of ATP release from the cytosol into the extracellular matrix in plants appeared earlier [12,13,14,15]. In plants, there are several possible ways of ATP outlet. The ATP release triggered by environmental stimuli appears via the wounded cell membrane [16], exocytosis [12], the p-glycoprotein (PGP1) belonging to the ATP-binding cassette ABC transporters [17], and plasma membrane-localized nucleotide transporters (PM-ANT1) [18]. The receptor-eATP interaction begins a cascade reaction that leads to further downstream physiological changes protecting the plant against both biotic and abiotic stresses but also guarantees proper plant growth and development [19].

In order to maintain proper cell growth and function, eATP concentration must be controlled. During regeneration after the stress factor has disappeared, eATP degradation is conducted by hydrolytic enzymes called apyrases [20]. The human apyrases are the best characterized and described apyrases among different kingdoms. Their cellular localization includes plasma membrane, Golgi apparatus, and endoplasmic reticulum. The human apyrases present in plasma membrane show ecto-apyrases activity mediating regulation of the eATP in the extracellular environment. The *Arabidopsis thaliana* apyrase family consists of seven enzymes among which two closely related ones, APY1 and APY2, are the most extensively characterized. These enzymes mediate the luminal glycosylation and can be a component of regulation of the eATP level. It was demonstrated that these two enzymes are an integral component of the Golgi apparatus membrane where they indirectly control the eATP level by modulating the luminal concentration of ATP in secretory vesicles [21,22,23] (Figure 1). Both APY1 and APY2 are also essential enzymes for proper plant growth and development. These processes are auxin-dependent. Among various factors, auxin transport also depends on the expression of the genes encoding APY1 and APY2. Suppression of the APY1 and APY2 expression causes dwarfism, impaired polar auxin transport and eATP over-accumulation in *Arabidopsis thaliana* [24,25]. However, it was suggested that some fractions of the APY1 and APY2 population with ecto-apyrases activity might by localized also in the plasma membrane [19]. Although there are many data regarding subcellular localization of *Arabidopsis* apyrases (APY1 and APY2), the apyrases from soybean (GS52), pea (PsAPY1), and potato (StAPY3) occur outside the cell (ecto-apyrases) [26,27,28]. A study conducted by Wu and co-workers indicated that the externally applied APY1 and APY2 inhibitors cause the increase of eATP and physiological changes typical for the plant reaction to stress [24]. Furthermore, there are some reports showing particular plant species secreting individual apyrase members out of the endomembrane system. Taking all these circumstances into account, the existence of a plasma membrane-localized *Arabidopsis thaliana* apyrase (Figure 1) cannot be excluded [29].

### 2.1. Plant eATP Receptors

The eATP animal receptors were discovered in 1976 [30]. Initially, they were called ‘purinergic receptors’. The name was changed to ‘nucleotide receptors’, as both purine and pyrimidine nucleotides trigger their activation [31]. These receptors belong to two groups, P1 and P2, but only P2 receptors are activated by ATP [32]. P2 receptors are divided into two classes: ligand-gated ion channels (P2X) and G protein-coupled (P2Y) receptors [33]. It is important to emphasize that P2X and P2Y receptors do not exist in plant organisms [34]. For many years it was hard to prove the existence of a similar receptor in plants although there were papers indicating changes in plants’ development, growth and response to stresses under exogenously applied ATP [11]. Research on *dorn1* (DOes not Respond to Nucleotides) mutants of *Arabidopsis thaliana* revealed the first plant receptor with a high affinity to bind eATP. Mutants showed a lower cytosolic Ca^2+^ level, lack of mitogen-activated protein kinases (MAPKs) activation and as a consequence decline of the defense-related gene expression level. Initially, the newly discovered receptor was named DORN1, but later the name was changed to P2K1 because the research showed that the *DORN1* gene encodes the L-type lectin receptor-like kinase I.9 (LecRK-I.9). This receptor has three domains: an extracellular ATP-binding lectin domain, a single transmembrane domain, and an intracellular kinase domain (Figure 1). In the case of *dorn1* mutants defective in the kinase domain, eATP is not able to connect with the receptor and as a result is not able to trigger downstream responses. Decreased expression of the *LecRK-I.9* gene caused weaker eATP responses, while intensified *LecRK-I.9* gene expression enhanced eATP responses [35]. DORN1 is a plant purine receptor that belongs to the lectin-receptor kinase family and is denoted as P2K1 to distinguish it from the animal P2 receptors: P2X and P2Y [11,36,37]. Interestingly, *P2K1* gene expression level is relatively high during the major stages of plant development [37], suggesting the essential role of the eATP in processes such as seedling growth, stomata movement, pollen tube development, root hair growth, gravitropism, and biotic and abiotic stress responses [38].

Recently, the existence of another, unidentified, non-P2K1 receptor with affinity to eATP was suggested [39]. It was reported that *Arabidopsis thaliana dorn1* null mutants demonstrated an increased level of cytosolic Ca^2+^ under the exogenously applied ATP. It was a result of the heterotrimeric G-protein G_α_ subunit activation followed by the opening of a plasma-membrane Ca^2+^ channel. This finding suggests new, P2K1-independent responses to eATP, involved among others in the root-bending mechanism [39]. Moreover DORN1 could underpin several calcium-related responses but it may not be the only receptor for eATP in *Arabidopsis thaliana* [40].

### 2.2. Plant eATP Signal Transduction Pathway

In plants, eATP takes part in cell signaling as a messenger, which triggers a signaling cascade, when binding to the P2K1 receptor (Figure 1). Cytosolic nitric oxide (NO), Ca^2+^, and ROS form a secondary messenger trio, which occurs in various signaling pathways leading to the transient phosphorylation of MAPK, especially MPK3 and MPK6, and expression of defense-related genes [41]. The multibranched reaction starts with the elevation of the cytosolic Ca^2+^ level triggered by eATP, leading to the accumulation of MAPK and NO, as well as to the phosphorylation and activation of the RBOHD (respiratory burst oxidase homolog protein D) subunit of the plasma membrane-localized NADPH oxidase [42]. This enzyme catalyzes the synthesis of extracellular ROS like superoxide (O^2−^), which is then converted into hydrogen peroxide (H_2_O_2_) in the extracellular milieu [16,33,43]. Extracellular H_2_O_2_ crosses the double-layered plasma membrane via various channels among which aquaporins are distinguishable [44]. Subsequently, ROS trigger changes in the expression of nuclear genes, responsible for defense responses [16], root hair growth [12] or regulation of Na^+^, H^+^, and K^+^ levels. Consequently, a disturbance in the ion homeostasis may lead to the mitochondria-independent type of programmed cell death by activation of the caspase-like proteases [45].

### 2.3. eATP Involvement in Plant Responses to Biotic and Abiotic Stresses

Aside from different eATP-induced factors and signaling molecules contributing to the plant resistance against pathogens and various abiotic stresses, classical defense hormones such as jasmonate, ethylene, and salicylic acid are of high importance [38]. Based on the pathogens’ lifestyles two groups of these organisms are distinguishable: biotrophs and necrotrophs. Interestingly, the efficiency of the used phytohormone against these two groups of pathogeneses differs. Salicylic acid induces plant defense against biotrophic pathogens, whereas jasmonate and ethylene are indispensable when necrotrophic pathogens and herbivorous insects attack the plant. Furthermore, salicylic acid regulates pathogen-induced systemic acquired resistance (SAR), whereas jasmonate and ethylene play a crucial role in rhizobacteria-mediated induced systemic resistance (ISR) [46]. Transcriptomic research of *Arabidopsis thaliana* mutants defective in the jasmonate, ethylene, and salicylic acid signaling pathway, which were treated with ATP, revealed crosstalk in the signaling of the typical plant hormones and eATP. These results showed that from among all of the defense-related genes, up to 50% were induced by eATP in cooperation with the typical plant defense hormones. This finding suggests a complex network of the plant defense mechanism that needs to be explored more deeply at different levels [38]. The contribution of transcription factors to ATP-responsive transcription is also considered. It was demonstrated that a calmodulin-binding transcription activator (CAMTA3) and MYC transcription factors are required for proper defense-related gene transcription, whose expression is induced by eATP [38].

In order to examine the importance of eATP in the plant reaction to pathogens two approaches might be considered: either alteration of the P2K1 receptor or manipulation at the eATP level. Overexpression of the P2K1 gene resulted in increased plant resistance to the bacterial pathogen *Pseudomonas syringae* and the oomycete pathogen *Phytophthora brassicae* [47,48]. Plants treated with ATP were found to be protected against various organisms such as the fungal pathogen *Botrytis cinerea* [7] and the bacterial pathogen *Pseudomonas syringae* [48]. Although the complete mechanisms of eATP signal transduction through the P2K1 receptor remain unclear, a plenitude of evidence shows the involvement of eATP in plant resistance to biotic stresses.

Studies show that eATP level, as well as P2K1 activity, are also of high importance in the plant-fungus symbiosis. Interactions of the filamentous root endophyte *Serendipita indica* with various experimental plant hosts, including *Arabidopsis thaliana* and *Hordeum vulgare*, have been examined. *Serendipita indica* colonization was found to be beneficial for plants, reflected in plant growth enhancement, assimilation of nitrate and phosphate improvement and better tolerance to both abiotic and biotic stresses [49]. Although *Serendipita indica* penetrates the host’s root cells, massive plant cell death does not occur. It is because of an enzymatically active ecto-5′-nucleotidase (E5′NT) enzyme secreting by *Serendipita indica* which is capable of hydrolyzing nucleotides in the apoplast. By the hydrolyzation of ATP, ADP, and AMP, *Serendipita indica* E5′NT modifies the eATP level, leading to switching off the plant defense reactions which promote proper fungal accommodation [50]. Moreover, it was observed that the rhizobial nodule factor stimulates the release of ATP outside the cell by the root hair tips of *Medicago truncatula* [12]. Treatment of plants with eATP caused also the change their susceptibility to pathogen infection [16]. It is hypothesized that ecto-apyrases may decrease the eleveted level of eATP upon symbiont infection and thereby prevents the activation of plant defense pathways that could limit symbiont invasion [51].

Several studies indicate the role of eATP in the response to different types of abiotic stresses. In addition to mechanical stimuli caused by wounding or touch, ATP is also released in response to treatment with molecules such as abscisic acid and L-glutamate [15,52]. A similar reaction was observed during both osmotic and salt stress [12,15,53] as well as under cadmium treatment [54]. Consequently, eATP accumulation triggers plant physiological changes leading to enhancement of resistance. It involves rapid closure of leaf stomata [55], probable seedling viability enhancement [56] and modification of root growth direction when encountering an obstacle [34]. In addition, hypertonic salt stress interferes with the photosynthesis machinery by decreasing the levels of maximal efficiency of photosystem II and depleting the photochemical quenching [54]. Moreover, abiotic stress caused by cadmium triggers a rapid increase in lipid peroxidation but also higher antioxidant and lipoxygenase activities in *Arabidopsis thaliana* cells [54]. These reactions are associated with boosted synthesis of jasmonic acid, which is one of the most important molecules involved in the different stress responses [57]. Nowadays, one of the major anthropogenic pollutants with the highest level of threat to human health is cadmium [58]. It is also considered as one of the most phytotoxic elements among heavy metals, causing a reduction in crop biomass by disrupting photosynthesis and respiration [59]. In addition, cadmium can induce other physiological changes including induction of oxidative stress by boosting the production of ROS [58], modifications in gene expression [60,61], as well as changes in enzyme activity [62], hence activating plant defense.

## 3. Extracellular Pyridine Nucleotides

The pyridine nucleotides nicotinamide adenine dinucleotide (NAD^+^) and NAD^+^ phosphate (NADP^+^) are commonly occurring electron carriers that are involved in metabolic reactions as well as intracellular signaling [63,64]. It is known that in plants NAD(P)^+^ can be released outside the cell [65]. Extracellular NAD(P)^+^ (eNAD(P)^+^) induces the expression of pathogen-related genes and the resistance to *Pseudomonas syringae* in *Arabidopsis thaliana* through pathways involving calcium- and salicylic acid-mediated defense signaling. It was also indicated that eNAD(P)^+^ induces transcriptional and metabolic changes in *Arabidopsis thaliana* similar to those caused by pathogen infection [66]. Moreover, the expression of the human NAD(P)^+^-hydrolyzing ecto-enzyme CD38 in *Arabidopsis thaliana* partially compromises systemic acquired resistance (SAR) and this suggests that eNAD(P)^+^ can be a SAR signal molecule [67]. Recent studies on the function of eNAD(P)^+^ focus on understanding their role in plants and on identifying their receptor(s). Analysis of transcriptome changes in *Arabidopsis thaliana* evoked by eNAD^+^ identified a lectin receptor kinase (LecRK) LecRK-I.8 as a potential eNAD^+^ receptor. LecRK-I.8 is located in the plasma membrane, has kinase activity, and specifically binds only NAD^+^, but not NADP^+^, ATP, ADP or AMP. Moreover, the expression of LecRK-I.8 is induced by the eNAD^+^ [68]. Another LecRK that can be a potential receptor for eNAD(P)^+^ is LecRK-VI.2, which was identified in *Arabidopsis thaliana*. LecRK-VI.2 is constitutively associated with Brassinosteroid Insensitive1-Associated Kinase1 (BAK1) and it was shown that complex LecRK-VI.2/BAK1 is involved in SAR [69].

## 4. Uncommon Nucleotides as Signaling Molecules

### 4.1. Mononucleoside Polyphosphates

#### 4.1.1. Structure and Occurrence of Mononucleoside Polyphosphates

Despite mono- and dinucleoside polyphosphates having been discovered in the middle of the twentieth century, our knowledge about their biological function is still poor, especially in plants. Mononucleoside polyphosphates (p_n_Ns) contain a nucleoside and oligophosphate chain. Examples of these nucleotides are adenosine 5′-tetraphosphate (p_4_A, ppppA, Figure 2) and adenosine 5′-pentaphosphate (p_5_A, pppppA). Both of them were discovered in commercial preparations of ATP obtained from bovine cells [70,71], horse muscle [72] and yeast [73]. Additionally, other purine and pyrimidine p_4_Ns were found as contamination of various nucleoside triphosphates (NTP) preparations: p_4_G [74,75], p_4_U [76], and p_4_C [77]. The existing of p_4_A and p_5_A was confirmed in biological materials such as rat liver [78,79], rabbit and horse muscle [80], bovine adrenal medulla [80,81,82,83], rabbit thrombocytes [84], and *Saccharomyces cerevisiae* [85]. However, the concentration of p_4_A in the above-mentioned animal samples was about 2 μM, but in chromaffin granules from the adrenal medulla it was about 800 μM, and it was even up to 4 orders of magnitude and 300-fold lower than ATP concentration, respectively [83]. Until now there is no information about the content of p_n_Ns in the plant tissues.

#### 4.1.2. Synthesis and Degradation of Mononucleoside Polyphosphates

The level of p_n_Ns in a cell depends on its biosynthesis and degradation, but also p_n_Ns may be a product of the degradation of some dinucleoside polyphosphates (Np_n_N’s). Enzymes that can synthesize p_n_Ns in vitro can be grouped into two categories: aminoacyl-tRNA synthetases (AARS) and non-aminoacyl-tRNA synthetases (non-AARS) [86]. Among enzymes synthesizing p_n_Ns only one belongs to the AARS and it is lysyl-tRNA synthetase (LysRS) from *Escherichia coli*, which can synthesize p_4_A [87,88]. The majority of the enzymes belong to the non-AARS. Table 1 presents non-plant enzymes synthesizing p_n_Ns.

Among enzymes that can degrade p_n_Ns some of them exhibit low substrate specificity, for example, alkaline (EC 3.1.3.1) and acid phosphatases (EC 3.1.3.2), that release phosphate residues up to adenosine [113]. Apyrase (EC 3.6.1.5) also can cut off phosphate residues, but only to AMP [113]. Phosphodiesterase I (EC 3.1.4.1) degrades p_4_A to triphosphate and AMP [100]. Additionally, adenosine-phosphate deaminase from *Aspergillus oryzae* and *Helix pomatia* is involved in p_4_A metabolism, converting it into inosine 5′-tetraphosphate (p_4_I) [114].

As mentioned above, p_n_Ns can accumulate in the cell as a result of the degradation of Np_n_N’s. The way in which p_4_A can accumulate in a cell is the degradation of diadenosine 5′,5′′′-pentaphosphate (Ap_5_A) and diadenosine 5′,5′′′-hexaphosphate (Ap_6_A). The degrading enzymes include phosphodiesterase I (EC 3.1.4.1) occurring in prokaryotes and eukaryotes [115,116], symmetrical dinucleoside tetraphosphatase (EC 3.6.1.41) from bacteria [117], dinucleoside tetraphosphate phosphorylase (EC 2.7.7.53) from *Saccharomyces cerevisiae* and *Euglena gracilis* [118,119], and dinucleoside triphosphatase (EC 3.6.1.29) among others from *Lupinus luteus* [120].

Despite the lack of knowledge about the occurrence and the concentration of mononucleoside polyphosphates in higher plants, there are described a few enzymes which can synthesize p_n_Ns. All of them are listed in Table 2. The first described plant enzyme that synthesizes p_n_Ns is 4-coumarate:CoA ligase (4CL2) from *Arabidopsis thaliana* that catalyzes the reaction of the synthesis of both p_4_A and p_5_A [121]. This enzyme is a branch point in the phenylpropanoid pathway that leads to the biosynthesis of flavonoids, lignin, and stilbenes. It is known that the phenylpropanoid pathway is involved in plant responses to numerous environmental stimuli, especially under biotic and abiotic stresses [122]. Another enzyme synthesizing p_4_A is jasmonate:amino acid synthetase from *Arabidopsis thaliana* (JAR1) [123]. JAR1 is involved in the function of jasmonic acid (JA) as a plant hormone and it catalyzes the synthesis of several JA-amido conjugates, the most important of which appears to be jasmonic acid-isoleucine. Both of the above-mentioned plant enzymes belong to the acyl~adenylate-forming firefly luciferase superfamily [124] which catalyzes a two-step reaction. During the first step, acyl and ATP form an acyl-adenylate intermediate with the simultaneous release of pyrophosphate (PP_i_). In the second step, 4CL in the absence of CoA catalyzes the formation of uncommon mononucleoside polyphosphates such as p_4_A and p_5_A [121].

In plants there also exists an enzyme degrading p_4_A. It is nucleoside tetraphosphate hydrolase (EC 3.6.1.14) occurring in *Lupinus luteus* seeds. This enzyme hydrolyzes both p_4_A and p_4_G with the same rate while p_5_A is degraded up to 200-fold more slowly than p_4_A and p_4_G [120].

### 4.2. Adenosine 5′-Phosphoramidate

#### Structure, Occurrence, and Metabolism of Adenosine 5′-Phosphoramidate

Among naturally occurring uncommon mononucleotides is adenosine 5′-phosphoramidate (NH_2_-pA, Figure 3). This compound is believed to occur in all organisms; however, so far NH_2_-pA has only been detected among cellular nucleotides purified from the green alga *Chlorella pyrenoidosa* [125]. Similarly to p_n_Ns, the level of NH_2_-pA in a cell is enzymatically controlled. It is known that NH_2_-pA can be synthesized by adenylyl sulfate:ammonia adenylyltransferase (EC 2.7.7.51) in the algae *Chlorella pyrenoidosa*, *Euglena gracilis*, amoeba *Dictyostelium discoideum*, bacteria *Escherichia coli*, and in higher plants *Hordeum vulgare*, *Spinacia oleracea* [126], and *Lupinus luteus* [127]. This transferase catalyzes the following reaction: SO_4_-pA + NH_4_^+^ → NH_2_-pA + SO_4_^2−^ + 2H^+^.

The supposition that NH_2_-pA is a ubiquitous compound and that its concentration is enzymatically controlled may be supported by the existence of various enzymes that catalyze the cleavage of NH_2_-pA to ammonia and AMP (pA) by hydrolysis [128,129,130,131,132,133], or to ammonia and ADP (ppA) by phosphorolysis [130]. Both the synthesis [127] and the degradation [131] of NH_2_-pA can be controlled by HIT (histidine triad) proteins. One of the proteins belonging to the HIT family proteins is Fhit (fragile histidine triad). Based on the mechanism of the action of Fhit protein, which is able to hydrolyze the P-N bond in adenosine phosphoimidazolide [134], it is hypothesized that other uncommon nucleotides can be substrates for human and *Arabidopsis thaliana* Fhit. Other proteins belonging to HIT family proteins are Hint protein (having the activity of NH_2_-pA hydrolase) and GalT proteins (having the specific activity of nucleoside monophosphate transferase) [135]. The protein Fhit from human and *Arabidopsis thaliana* exhibited the activity attributed to adenylyl sulfate sulfohydrolase and nucleoside phosphoamidases releasing AMP from adenosine 5′-phosphosulfate (SO_4_-pA) and NH_2-_pA, respectively. Fhit protein also catalyzes the hydrolysis of the P-F bond in the synthetic nucleotide adenosine 5′-monophosphofluoride (F-pA) releasing AMP [136]. Recently it has been shown that NH_2_-pA can also be synthesized by Fhit proteins from *Lupinus luteus* seeds and *Arabidopsis thaliana.* It catalyzes the ammonolysis of SO_4_-pA, leading to the formation of NH_2_-pA [127].

### 4.3. Dinucleoside Polyphosphates

#### 4.3.1. Structure and Occurrence of Dinucleoside Polyphosphates

Dinucleoside polyphosphates (Np_n_N’s) consist of an oligophosphate chain that links the two 5′-esterified nucleosides. Among Np_n_N’s that are most frequently tested are adenine nucleotides: diadenosine triphosphate (Ap_3_A) and diadenosine tetraphosphate (Ap_4_A, Figure 4). The first reports about the existence of Np_n_N’s come from the 1950s. It was discovered that during the chemical synthesis of adenosine-uridine monophosphate (ApU) there were synthesized byproducts such as diadenosine diphosphate (Ap_2_A) and diuridine diphosphate (Up_2_U) [137]. Nevertheless, the first Np_n_N discovered in the biological material was diguanosine tetraphosphate (Gp_4_G). The presence of Gp_4_G and Gp_3_G was observed in biological preparations in encysted gastrulae of the brine shrimp and *Artemia salina* in millimolar concentrations [138,139]. The most common Np_n_N is Ap_4_A, which for the first time was detected in rat liver, and its concentration was estimated at 30 nM [97]. Subsequently Ap_4_A and other Np_n_N’s have been observed in submicromolar concentrations in many investigated cells and tissues [140,141]. Np_n_Ns were identified in bacteria [142,143], yeast [144,145], and animal and human cells [140,146,147,148,149,150]. So far, Np_n_N’s have not been identified in plants.

#### 4.3.2. Synthesis and Degradation of Dinucleoside Polyphosphates

The level of Np_n_N’s in cells may be a result of the synthesis and degradation of these compounds. Table 3 lists non-plant enzymes synthesizing various Np_n_N’s. Enzymes synthesizing Np_n_N’s are some ligases [89,92,106,121,151,152], luciferase from *Photinus pyralis* [153], and some transferases [100,119,154]. Among aminoacyl-tRNA synthetases, the most effective in Ap_n_N synthesis are lysyl- (EC 6.1.1.6), phenylalanyl- (EC 6.1.1.20), alanyl- (EC 6.1.1.7), and prolyl- (EC 6.1.1.15) t-RNA synthetases [86]. Moreover, the RNA-dependent RNA polymerase elongation complex from hepatitis C virus (HCV) can use in vitro nucleoside triphosphates (NTPs) to excise the terminal nucleotide in nascent RNA and mismatched ATP, UTP, or CTP could mediate excision of 3′-terminal CMP to generate the dinucleoside tetraphosphate products Ap_4_C, Up_4_C, and Cp_4_C, respectively [155].

So far, only three enzymes synthesizing Np_n_N’s have been described in plants (Table 4). Among aminoacyl-tRNA synthetases only phenylalanyl- (EC 6.1.1.20) and seryl-tRNA (EC 6.1.1.11) synthetases from *Lupinus luteus* [152] synthesize Ap_n_N. The mechanism of the synthesis of Ap_n_Ns catalyzed by plant aminoacyl-tRNA synthetases is based on the formation of aminoacyl~pA and the transfer of adenylate to pppN [152]. The third enzyme which is involved in the synthesis of Ap_n_N in plant cells is 4:coumarate-CoA ligase (EC 6.2.1.12) [121].

Enzymes degrading Np_n_N’s may be divided into substrate-specific and substrate non-specific. Substrate-specific enzymes degrading Np_n_N’s include, among others, *asymmetrical* Np_4_N hydrolase (EC 3.6.1.17), *symmetrical* Np_4_N hydrolase (EC 3.6.1.41), and dinucleoside triphosphate (Np_3_N) hydrolase (EC 3.6.1.29) [156]. In plants, the activity of *asymmetrical* Np_4_N hydrolase was detected in *Lupinus luteus* seeds [157], *Helianthus annuus* and marrow (*Cucurbita pepo*) seeds [158], tomato cells [159], *Lupinus angustifolius* seeds [160] and in *Hordeum vulgare* [161]. Among Np_4_N’s, only Ap_4_A can be degraded by the *asymmetrical* Np_4_N hydrolase giving ATP. The substrate non-specific enzymes in plants are phosphodiesterase I (EC 3.1.4.1) from *Lupinus luteus* [157] and nucleotide pyrophosphatase (EC 3.6.1.9) from potato tuber [162].

### 4.4. Function of Uncommon Nucleotides

The cellular level of Np_n_N’s is modified under various physiological and pathological conditions and the compounds have been suggested as intracellular messengers in diverse cellular processes [163,164]. The increased concentration of Ap_4_A was correlated with high cellular proliferation rates [165] or some phases of the cell cycle [165,166]. A dramatic increase of levels of Np_4_N and various Np_n_N’s has been observed in cells subjected to stresses, such as elevated temperature, ethanol or cadmium [143,144,145,167]. A very interesting function of uncommon nucleotides is their potential function as alarmones [168]. However, so far no clear metabolic or molecular targets or receptors of the postulated alarm signaled by the Np_n_N’s have been experimentally demonstrated. Alarmones are intracellular signaling molecules that are produced under adverse environmental factors. They regulate gene expression at the transcriptional level. Nucleotides (p)ppGpp and ppGpp have such a role in bacteria [169]. The results of recent studies on the role of uncommon nucleotides focus on microorganisms and animals, including man. It has been demonstrated that Np_n_N’s may accumulate in cells under stress conditions. In cells of *Saccharomyces cerevisiae* and *Escherichia coli* under elevated temperature and cadmium ions an increased level of Np_n_N’s was observed. The intracellular concentrations of these molecules vary from 10^−9^ M in a basal metabolic state to 10^−4^ M when the organisms are subjected to stresses, such as heat shock or exposure to cadmium [143,167]. In cyanobacteria the same effects are a result of thermal shock and heavy metals [145]. In *Salmonella*, the increased synthesis of Np_n_N’s was caused by ethanol [142]. Studies performed on orange fruit showed that under high temperature enzymes involved in Np_n_N’ metabolism were activated and an extremely large increase in bis(5′-adenosyl) triphosphatase protein content was observed. These results suggest intensive synthesis of Ap_3_A under high temperature [170]. Another group of enzymes involved in Np_n_N’ metabolism is Nudix (nucleoside diphosphate linked to x) hydrolases (NUDTs). An increase of transcript and protein NUDT7 levels was found in *Arabidopsis thaliana* plants under biotic stress evoked by *Pseudomonas syringae* avrRpt2 [171]. In *Arabidopsis thaliana*, there was also observed a quick increase in NUDT7 content under the effect of ozone and pathogen infection [172]. Recent studies on AtNUDT19 indicate the involvement of this hydrolase in the photo-oxidative stress response by regulating photosynthesis, the antioxidant system and the synthesis of salicylic acid—a signaling molecule involved in the response to biotic and abiotic stresses [173]. Additionally, it is suggested that barley NUDTs respond to abiotic stress [174]. It is also known that NUDTs bind RNA and participate in the regulation of gene expression in animals and plants [175,176,177].

Studies on the function of Ap_4_A in *Escherichia coli* have shown that it may be a damage metabolite and a few proteins binding Ap_4_A in these bacteria were identified [178]. Recent studies on the function of Ap_4_A have shown that aminoglycoside antibiotics induced synthesis of Ap_4_A in *Escherichia coli* and it caused bacterial cell killing by these antibiotics [179]. Lysyl-tRNA synthetase plays a key role in MITF (microphthalmia-associated transcription factor) transcriptional activity via Ap_4_A as an important signaling molecule in mast cells [180,181]. Studies on chronic myelogenous leukemia cells showed that NUDT2 disruption elevates Ap_4_A and down-regulates immune response and cancer promotion genes [182]. Ap_3_A and Ap_4_A induce activation of a signaling pathway that results in increase of proliferation of vascular smooth muscle cells by stimulation of ERK1/2(MAP kinase) [183]. It is known that Np_n_N’s can act as neurotransmitters and influence the vascular system through purinergic receptors. There is also evidence that Ap_4_A inhibits the initiation of DNA replication. It is also proposed that Ap_4_A acts as an important inducible ligand in the DNA damage response to prevent the replication of damaged DNA [184]. In contrast, the increase in the ratio of Ap_3_A/Ap_4_A in human HL60 cells can induce apoptosis [164].

So far there are only a few papers describing the role of uncommon nucleotides in plant cells. It was shown that the phenylpropanoid pathway may be affected by exogenous uncommon nucleotides [185,186,187,188]. The phenylpropanoid pathway is a source of secondary metabolites which are very important in plant responses to biotic and abiotic stresses [122]. The first data showed that Ap_3_A and Ap_4_A regulate the expression of genes and the activity of enzymes involved in the phenylpropanoid pathway in seven-day old *Arabidopsis thaliana* seedlings [185]. Ap_3_A and Ap_4_A caused an increase in the activity of phenylalanine ammonia-lyase (PAL) and 4:coumarate-CoA ligase (4CL) just 10 min after application. Analysis of *PAL* gene expression showed that there was strong induction (about 70-fold) of *PAL2* gene expression by Ap_3_A and Ap_4_A. Additionally, PAL activity was significantly enhanced by Ap_3_A (about 9-fold). Moreover, it has been shown that the activity of 4CL and the *4CL* gene expression level were higher in seedlings treated with Ap_3_A and Ap_4_A [185]. These results may indicate a dual role of 4CL in the plant response to stress factors. Firstly, this enzyme synthesizes Np_n_N’s, i.e., compounds that can play a function as alarmones [121], and secondly, it synthesizes secondary metabolites minimizing the adverse effects of stresses in plant cells [189]. Also, Ap_3_A caused changes in the gene expression level and the activity of chalcone synthase (CHS) in *Arabidopsis thaliana* seedlings. This enzyme catalyzes the synthesis of chalcone, which is the precursor of secondary metabolites—flavonoids [185]. These data strongly indicate the signaling role of Np_n_N’s in plants. The inductive effect of Ap_3_A on gene expression of the phenylpropanoid pathway proteins has also been confirmed in a cell suspension of *Vitis vinifera* cv. Monastrell. A synergistic effect evoked by Ap_3_A and cyclodextrins in *trans*-resveratrol biosynthesis has been demonstrated [186]. It may suggest the involvement of Np_n_N’s in the plant response to stress. Interestingly, cyclodextrins act as elicitors in their chemical similarity to the alkyl-derivatized pectic oligosaccharides, which are released from the cell wall during fungal infection [190]. It is known that one of the defense strategies of higher plants against biotic and abiotic stresses is activation of the phenylpropanoid pathway leading to enhanced production of various phenylpropanoid compounds, such as flavonoids [191,192], lignin [193,194], anthocyanins [194,195,196], and salicylic acid [197] (Figure 5). Recently the differences in the regulation of gene expression of the phenylpropanoid pathway enzymes and phenylpropanoid accumulation by purine or pyrimidine Np_n_N’s in *Vitis vinifera* suspension cell culture were described. The pyrimidine dinucleotides, such as Cp_3_C, Cp_4_C, and Ap_4_C, markedly (6- to 8-fold) induced the expression of the gene coding the cinnamoyl-CoA:NADP oxidoreductase (CCR) that controls lignin biosynthesis. The most effective in stilbene accumulation was Up_4_U, but other pyrimidine dinucleotides (Cp_3_C, Cp_4_C, and Ap_4_C) strongly inhibited the biosynthesis of these phenylpropanoids [188].

As described above, in plants, the presence of enzymes degrading and synthesizing another uncommon nucleotide, NH_2_-pA, has been described [127,132]. It was evidenced that NH_2_-pA regulates the expression of genes coding enzymes of the phenylpropanoid pathway, such as PAL, cinnamate-4-hydroxylase (C4H), 4CL, CHS, CCR, and isochorismate synthase (ICS) in *Arabidopsis thaliana* seedlings. CCR is an enzyme involved in the biosynthesis of lignin, whereas ICS is involved in the biosynthesis of a signaling molecule—salicylic acid. Among the analyzed genes the strongest induction in gene expression caused by NH_2_-pA was observed for *CCR2* (approx. 4-fold). This was also accompanied by an increased level of lignin in the seedlings. Another important effect caused by NH_2_-pA was an increase in the *ICS* gene expression and a significant increase (2-fold) in the level of free salicylic acid. In view of the fact that salicylic acid is one of the signaling molecules involved in the response of plants to biotic stress, the induction of the synthesis of salicylic acid by NH_2_-pA may suggest the interaction of both these compounds in plant responses to stresses [187].

Another uncommon nucleotide occurring in plants is nicotinic acid adenine dinucleotide phosphate (NAADP). The synthesis of NAADP is not fully understood. However, it is supposed that it can be synthesized by adenosine-5′- diphosphateribosyl-cyclase (EC 3.2.2.5) [198]. This nucleotide is involved in calcium signaling in many organisms including plants [199,200,201]. NAADP-mediated calcium release has been shown in the microsomal vesicles of red beets (*Beta vulgaris*) and cauliflower (*Brassica oleracea*). Analysis of sucrose gradient-separated cauliflower microsomes revealed that the NAADP-sensitive Ca^2+^ pool was derived from the endoplasmic reticulum [199].

### 4.5. Uncommon Nucleotide Signal Transduction Pathways in Plants

As yet there has been no information about signal transduction pathways or receptors activated by uncommon nucleotides. Based on the data existing in the literature about the involvement of these molecules in different biochemical and physiological processes in bacteria, fungi, animal, and plant organisms, we propose putative signaling pathways induced by uncommon nucleotides in plant cells (Figure 5). It is hypothesized that uncommon nucleotides may be signal molecules synthesized by pathogens outside the cell and interact with plant cells through unknown plasma membrane receptor(s). These nucleotides can also be transported inside the plant cell by unknown plasma membrane transporter(s). In addition, their level in the plant cell can be regulated by synthesizing and degrading enzymes. Both the extracellular and intracellular uncommon nucleotides may affect the biosynthesis of other plant cell signal molecules, e.g., secondary messengers and hormones that can engage MAPK cascades, which are involved in plant growth and development, cellular responses to hormones, regulation of the cell cycle, and responses to biotic and abiotic stresses, such as pathogen infection, wounding, low temperature, drought, high salinity, metals, and ROS. Moreover, it is known that MAPK cascades can regulate gene transcription by activation or repression of transcription factors. In addition, MAPK transcript levels in *Arabidopsis thaliana* seedlings were shown to increase in a time-dependent manner following exposure to Cu and Cd [202]. Many MAPK cascades respond to hormones such as abscisic acid, jasmonic acid, salicylic acid, ethylene, auxins, and brassinosteroids. Usually, signaling molecules participate in distinct signaling pathways, resulting in the formation of a cross-talking network that co-ordinates responses to different stresses [202,203,204]. We postulate that MAPK cascades can be involved in the signal transduction pathway triggered by uncommon nucleotides. So far, it is only known that extracellular uncommon nucleotides can induce gene expression of the phenylpropanoid pathway, the accumulation of phenylpropanoids, and the synthesis of salicylic acid. However, components of the nucleotide transduction pathway are still not known (Figure 5).

## 5. Conclusions

Plants are not able to escape from their abiotic (cold, heat, etc.) or their biotic (herbivores, insects) environment [205]. Furthermore, their food uptake and gas exchange take place through external surfaces (leaves: light and CO_2_, roots: ions and water). Plants must, therefore, possess systems to exchange information throughout the entire plant to ensure the coordination of plant development and defense. Evidence suggests that in plants information exchange relies on at least two different systems: one involving molecules that are transported within the plant and another that uses electrical and/or hydraulic signals to carry the information throughout the entire organism [205]. The systems for transmitting this information are complex and involve multiple components, which are far from being understood. For many years plant hormones were considered to be dominant molecules in plant signaling. Nowadays, this term is applied to many more compounds. It is suggested that dynamic changes in the level of second messengers, such as Ca^2+^, ROS, and NO, serve as signatures for both intracellular signaling and cell-to-cell communications. These second messenger signatures work in concert with physical signal signatures (such as electrical and hydraulic signals) to create a “lock and key” mechanism that triggers an appropriate response to various stresses [206]. Part of the system, new players in plant signaling, are chemical signals such as extracellular ATP and uncommon nucleotides described and discussed in this work. These molecules are essential for proper coordination of processes in particular organs as well as responses to internal and external signals playing analogical or similar functions in plants and in animals. The data obtained during the last few years suggest that in plants both purine and pyrimidine dinucleotides should be considered as members of a family of naturally occurring stress signaling molecules. So far, our knowledge of these signals in plants is still insufficient to clearly understand their signaling role. The mechanisms of signal perception and transduction are unknown. Further studies are required, which should aim at describing the signaling networks involving uncommon nucleotides, and also perform omic experiments to identify the network components and processes that are regulated by these signals.

## Figures and Tables

**Figure 1 cells-09-00345-f001:**
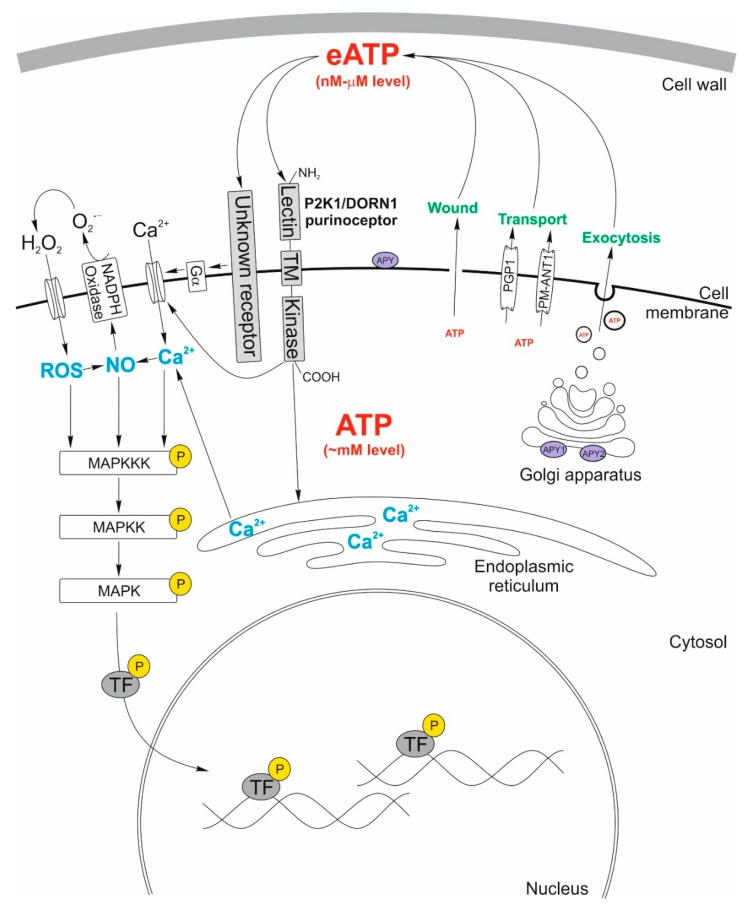
Model of changes occurring in the plant cell triggered by the extracellular ATP (eATP). In this model, three possible ways of ATP release into the extracellular matrix are demonstrated. It considers the wounded cell membrane, exocytosis, and two transporters: the p-glycoprotein (PGP1) belonging to the ATP-binding cassette ABC transporters, and the plasma membrane-localized nucleotide transporters (PM-ANT1). Two apyrases, APY1, and APY2, localized in the Golgi apparatus membrane of *Arabidopsis thaliana* regulate the concentration of eATP. Additionally, the hypothesized apyrase (APY) located at the extracellular surface of the plasma membrane can decrease eATP concentration directly in the extracellular matrix. The released eATP acts as a signaling molecule triggering elevation of the cytosolic Ca^2+^ level by activation of the P2K1 receptor, which in turn activates the Ca^2+^ channel. The hypothetical non-P2K1 receptor, whose binding with eATP leads to activation of the Gα subunit of the heterotrimeric G-protein, activates the cell membrane Ca^2+^ channel. High cytosolic Ca^2+^ concentration causes an increase in production of nitric oxide (NO), reactive oxygen species (ROS), and mitogen-activated protein kinases (MAPKs), which finally leads to various physiological responses. The ROS boosted production is due to the activation of the RBOHD subunit of the plasma membrane-localized NADPH oxidase. The contribution of the transcription factors in the regulation of gene expression is of high importance.

**Figure 2 cells-09-00345-f002:**
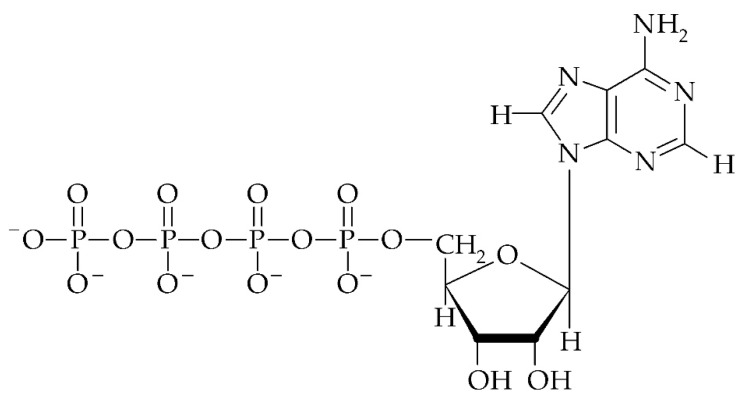
Structure of adenosine 5′-tetraphosphate (p_4_A).

**Figure 3 cells-09-00345-f003:**
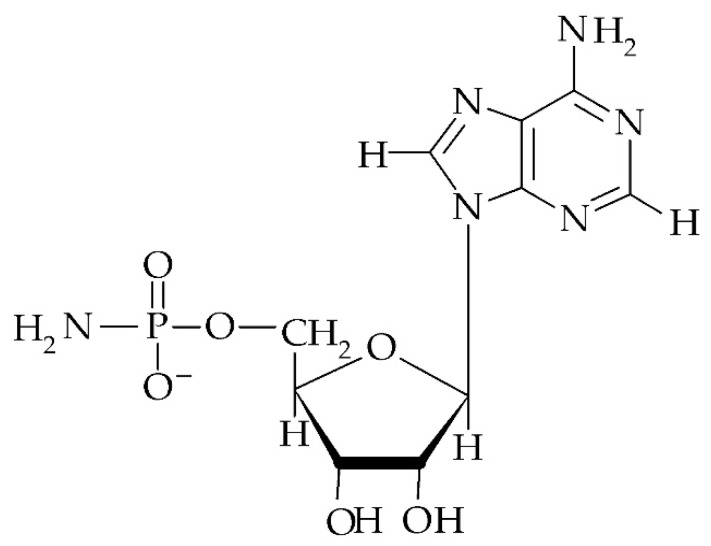
Structure of adenosine 5′-phosphoramidate (NH_2_-pA).

**Figure 4 cells-09-00345-f004:**
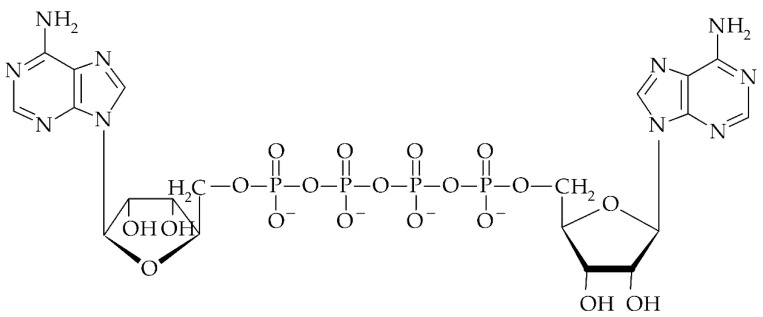
Structure of diadenosine 5′, 5′′′-tetraphosphate (Ap_4_A).

**Figure 5 cells-09-00345-f005:**
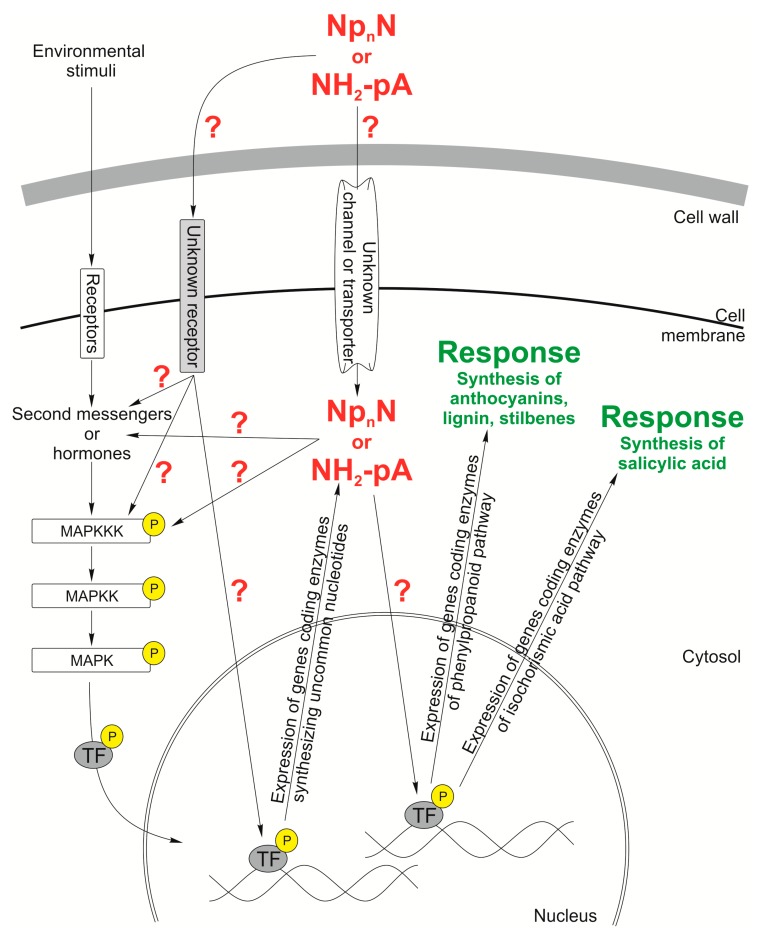
Hypothetical working model of Np_n_N’ and NH_2_-pA signaling network in a plant cell. Dinucleoside polyphosphate (Np_n_N) and adenosine 5′-phosphoramidate (NH_2_-pA) trigger induction of the phenylpropanoid and the isochorismic acid pathways yielding metabolites protecting plant against various types of stresses. Plant cells respond to environmental stimuli by intensification of the production of various compounds, such as anthocyanins, lignin, stilbenes, and salicylic acid. Question marks indicate the hypothetical components of the signaling network.

**Table 1 cells-09-00345-t001:** Non-plant enzymes synthesizing mononucleoside polyphosphates (p_n_Ns).

Enzyme	Organism	Reaction (E, Enzyme)	References
Lysyl-tRNA synthetase (EC 6.1.1.6)	*Escherichia coli*	1st step: E + lysine + pppA ↔ ↔ E:lysyl~pA + PP_i_ 2nd step: E:lysyl~pA + ppp → → ppppA + lysine + E	[87,88,89,90,91,92,93,94,95,96,97,98]
Luciferase EC 1.13.12.7	*Photinus pyralis*	1st step: E + luciferin + pppA → → E-luciferin~pA + PP_i_ 2nd step: E-luciferin~pA + (p)ppp → → (p)ppppA+ luciferin + E	[99]
UTP:glucose-1-phosphate uridylyltransferase (EC 2.7.7.9)	Saccharomyces cerevisiae	1st step: glucose-1-P + pppU → → Upp-glucose + PP_i_ 2nd step: Upp-glucose + (p)ppp → → (p)ppppU + glucose-1-P	[100]
Phosphoglycerate kinase (EC 2.7.2.3)	*Saccharomyces cerevisiae *	1,3-ppGly + pppA ↔ ↔ 3-pGly + ppppA	[80,101]
Adenylate kinase (EC 2.7.4.3)	Rabbit and pig muscles	ppA + pppA ↔ pA + ppppA	[102]
Succinyl-CoA synthetase (EC 6.2.1.5)	*Escherichia coli*	E-P + pppA ↔ E + ppppA	[103]
Acetyl-CoA synthetase (EC 6.2.1.1)	*Saccharomyces cerevisiae*	acetyl~pA + ppp ↔ acetate + ppppA	[104,105]
Acyl-CoA synthetase (EC 6.2.1.3)	*Pseudomonas fragi*	acyl~pA + ppp ↔ fatty acid + ppppA	[106,107]
DNA ligase (EC 6.5.1.1)	T4 phage, *Pyrococcus furiosus*	E-pA + ppp ↔ E + ppppA	[105,108,109,110]
RNA ligase (EC 6.5.1.3)	T4 phage	E-pA + (p)ppp ↔ E + (p)ppppA	[111]
UDP-MurNAc-l-alanine:d-glutamate ligase (EC 6.3.2.9)	*Escherichia coli*	E:acyl~P + pppA ↔ ppppA + acyl + E	[112]

**Table 2 cells-09-00345-t002:** Plant enzymes synthesizing mononucleoside polyphosphates (p_n_Ns).

Enzyme	Organism	Reaction (E, Enzyme)	References
4-Coumarate:CoA ligase (4CL2) (EC 6.2.1.12)	*Arabidopsis thaliana*	1st step: E + coumarate + ATP → → E:coumaroyl~pA + PP_i_ 2nd step: E:coumaroyl~pA + (p)ppp → → (p)ppppA + coumarate + E	[121]
Jasmonate:amino acid synthetase (JAR1) (EC 6.3.2.52)	*Arabidopsis thaliana*	1st step: E + jasmonate + ATP → → E:jasmonyl~pA + PP_i_ 2nd step: E:jasmonyl~pA + ppp → → ppppA + jasmonate + E	[123]

**Table 3 cells-09-00345-t003:** Non-plant enzymes synthesizing dinucleoside polyphosphates (Np_n_N’s).

Enzyme	Organism	Reaction (E, Enzyme)	References
Luciferase (EC 1.13.12.7)	*Photinus pyralis *	1st step: E + luciferin + pppA → → E-luciferin~pA + PP_i_ 2nd step: E-luciferin~pA + pppN → → AppppN+ luciferin + E	[99]
GTP:GTP guanylyltransferase (EC 2.7.7.45)	*Artemia salina, Saccharomyces cerevisiae*	1st step: E + pppN ↔ E~pN + PP_i_ 2nd step: E~pN + pppN → NppppN′ + E	[154]
UTP:glucose-1-phosphate uridylyltransferase (EC 2.7.7.9)	*Saccharomyces cerevisiae*	1st step: glucose-1-P + pppU → → Upp-glucose + PP_i_ 2nd step: Upp-glucose + pppN → → UppppN + glucose-1-P	[100]
Acyl-CoA synthetase (EC 6.2.1.3)	*Pseudomonas fragi*	1st step: Fatty acid + pppA ↔ ↔ acyl~pA + PP_i_ 2nd step: acyl~pA + pppA → → AppppA + fatty acid	[106,107]
RNA ligase (EC 6.5.1.3)	T4 phage	E-pA + pppN ↔ E + AppppN	[111]
DNA ligase (EC 6.5.1.1)	T4 phage, *Pyrococcus furiosus*	E-pA + pppN ↔ E + AppppN	[105,108,109,110]
Lysyl-tRNA synthetase (EC 6.1.1.6)	*Escherichia coli *	1st step: E + lysine + pppA ↔ ↔ lysyl~pA + PP_i_ 2nd step: E:lysyl~pA + pppN → → AppppN + lysine + E	[86]

**Table 4 cells-09-00345-t004:** Plant enzymes synthesizing dinucleoside polyphosphates (Np_n_N’s).

Enzyme	Organism	Reaction (E, Enzyme)	References
Phenylalanyl-tRNA synthetase (EC 6.1.1.20)	*Lupinus luteus*	1st step: E + phenylalanine + pppA ↔ ↔ phenylalanyl~pA + PPi 2nd step: E:phenylalanyl~pA + pppN → → AppppN + phenylalanine + E	[152]
Seryl-tRNA synthetase (EC 6.1.1.11)	*Lupinus luteus*	1st step: E + serine + pppA ↔ ↔ seryl~pA + PPi 2nd step: E:seryl~pA + pppA → → AppppA + serine + E	[152]
4-Coumarate:CoA ligase (4CL2) (EC 6.2.1.12)	*Arabidopsis thaliana*	1st step: E + coumarate + pppA → → E:coumaroyl~pA + PPi 2nd step: E:coumaroyl~pA + pppN → → NppppN+ coumarate + E	[121]

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
