# Peer review of "New Insight into Plant Signaling: Extracellular ATP and Uncommon Nucleotides"

_cells, 2020, doi:10.3390/cells9020345_

Round 1

Reviewer 1 Report

The manuscript “New insights into Plant Signaling: Extracellular ATP and uncommon nucleotides” by Pietrowska-Borek and coworkers gives an overview about the current knowledge on the signaling role of eATP and uncommon nucleotides. This review is written by experts in the field of uncommon nucleotides and plant signaling. This manuscript describes the effects of eATP in detail. However, when compared to eATP, the signaling role of uncommon nucleotides in plant metabolism is barely known and rather new. During the past decade, the authors gained more and more evidence that uncommon nucleotides may also act as signaling compounds in plants. Not only the text but also the nice working model gives an idea of how these molecules may induce specific changes in plant metabolism, which finally result in the corresponding plant responses to various stressors. Generally, the manuscript is well written, it reports on an interesting and actual topic and provides new insights and ideas into plant metabolism, signaling, abiotic and biotic stress responses.

Major point: Although this review is quite comprehensive, I was missing at least some information about the signaling role of pyridine nucleotides (NAD, NADP) and the uncommon pyridine nucleotide (NAADP) in plants. I would recommend to include a special paragraph about pyridine nucleotides or to mention the signaling function of NAD and related players at least somewhere in the text and refer to the corresponding literature.

Minor points:. Lines 284-287: This order of listed organisms is confusing since it gives the idea that Dictyostelium and Escherichia represent algae. This should be corrected/clarified.

Author Response

Reviewer 1

The manuscript “New insights into Plant Signaling: Extracellular ATP and uncommon nucleotides” by Pietrowska-Borek and coworkers gives an overview about the current knowledge on the signaling role of eATP and uncommon nucleotides. This review is written by experts in the field of uncommon nucleotides and plant signaling. This manuscript describes the effects of eATP in detail. However, when compared to eATP, the signaling role of uncommon nucleotides in plant metabolism is barely known and rather new. During the past decade, the authors gained more and more evidence that uncommon nucleotides may also act as signaling compounds in plants. Not only the text but also the nice working model gives an idea of how these molecules may induce specific changes in plant metabolism, which finally result in the corresponding plant responses to various stressors. Generally, the manuscript is well written, it reports on an interesting and actual topic and provides new insights and ideas into plant metabolism, signaling, abiotic and biotic stress responses.

Major point: Although this review is quite comprehensive, I was missing at least some information about the signaling role of pyridine nucleotides (NAD, NADP) and the uncommon pyridine nucleotide (NAADP) in plants. I would recommend to include a special paragraph about pyridine nucleotides or to mention the signaling function of NAD and related players at least somewhere in the text and refer to the corresponding literature.

We would like not to describe the role of NAD and NADP because these are typical nucleotides, commonly occurring in plants and animals. The knowledge about their metabolism, function, and signaling is extensive, and the separate review paper could be written about these nucleotides. However, we added to the manuscript some data about extracellular NAD and NADP. We added a new section 3. Extracellular Pyridine Nucleotides and we described in this section the role of eNAD and eNADP only in plants (lines 226-244).

We also added a short paragraph in section 4.4. Function of Uncommon Nucleotides (lines 477-484) where we described the role of NAADP in plants. This paragraph is short because the majority of the knowledge about NAADP concerns animals (including humans). Since our manuscript is dedicated plants we decided not to add data about the role of NAADP in animals.

Minor points: Lines 284-287: This order of listed organisms is confusing since it gives the idea that Dictyostelium and Escherichia represent algae. This should be corrected/clarified.

We corrected this mistake. All groups of organisms are now precisely distinguished (line 314).

Reviewer 2 Report

The manuscript by Pietrowska-Borek is a well-written, hypothetical, theoretical article. Many purinergic scientists have been wondering about the uncommon nucleotides, especially ApnA, in plants. The manuscript describes how strikingly little is known on the accumulation and function (lines 400-445) of uncommon nucleotides in plants relative to other organisms. It is a very interesting, timely manuscript and suggests that more study in this area in plants could yield important discoveries. I have several comments below, most of which are just minor, except that the authors dismissed describing about extracellular pyridines as additional purinergic signal (see below in detail).

Page 2 Although there are many arguments regarding subcellular localization of Arabidopsis apyrases (APY1 and APY2), the apyrases from soybean (GS52), Pea (PsAPY1), and potato (StAPY3) are well established as an apoplastic enzymes, i.e., ecto-apyrase. The authors should mention about these apyrases by citing the following papers:

https://doi.org/10.1094/MPMI.2000.13.10.1053

https://doi.org/10.1007/s10327-006-0279-7

https://doi.org/10.1104/pp.108.117564

Line 132-133. The following paper should be added as a reference in which the transcriptome in “intensified LecRK-I.9 gene expression” was described.

https://doi.org/10.1080/15592324.2019.1659079

Line 146-150. Tanaka et al. 2014 is better to cite here since the paper summarized entire eATP signaling. https://dx.doi.org/10.3389/fpls.2014.00446

Paragraph 189-199. The author should be aware that eATP level also affect on infection of RN rhizobia and AM fungi. It would be nice to describe about it in the paragraph by citing the following paper: https://doi.org/10.1002/9781119053095.ch52

Lines 446-469 are highly speculative. It is interesting to think of these ideas for signal transduction by uncommon nucleotides, but I think the authors should place more stress on the highly speculative nature of the proposed pathways.

I would appreciate some discussion of the possibility that these uncommon nucleotides signal through their metabolic conversion to ATP. Does this idea make any sense? Does the extracellular ATP receptor mutant dorn1 still respond to these nucleotides?

It’s surprising that, while they describe extracellular ATP as a signal, they neglected to mention recent research on extracellular pyridine nucleotides, NAD(P), as signals. The authors should cover this topic in the manuscript by citing the relevant papers:

https://doi.org/10.1038/s41467-019-12781-7

https://doi.org/10.1111/j.1365-313X.2008.03687.x

https://doi.org/10.7554/eLife.25474

Line 463-464. "Many MAPK cascades respond to hormones such as abscisic acid, jasmonic acid, salicylic acid, ethylene, auxins, and brassinosteroids." I think the authors need to cite references for the relevant studies. For example, this paper shows that ABA or JA/coronatine treatment decreases the abundance of phosphorylated MAPKs after flg22 treatment by a transcriptional mechanism: https://www.pnas.org/content/early/2017/06/23/1702613114.

Is there any place for p4A to fit in the working model in Fig5?

Line 354-445. Some of the nucleotide effects seem as an intracellular function. The authors carefully describe all statements by specifying whether the effect is as an extracellular signal or an intracellular signal.

Line 483-486. The author should cite the following paper in which molecular signal signatures and physical signal signatures are well described and discussed.

https://doi.org/10.1016/j.plantsci.2019.03.004

Author Response

Reviewer 2

The manuscript by Pietrowska-Borek is a well-written, hypothetical, theoretical article. Many purinergic scientists have been wondering about the uncommon nucleotides, especially ApnA, in plants. The manuscript describes how strikingly little is known on the accumulation and function (lines 400-445) of uncommon nucleotides in plants relative to other organisms. It is a very interesting, timely manuscript and suggests that more study in this area in plants could yield important discoveries. I have several comments below, most of which are just minor, except that the authors dismissed describing about extracellular pyridines as additional purinergic signal (see below in detail).

According to the same requirement of Reviewer 1, we added appropriate fragments to the manuscript: section 3. Extracellular Pyridine Nucleotides (226-244) and paragraph in lines 477-484.

Page 2 Although there are many arguments regarding subcellular localization of Arabidopsis apyrases (APY1 and APY2), the apyrases from soybean (GS52), Pea (PsAPY1), and potato (StAPY3) are well established as an apoplastic enzymes, i.e., ecto-apyrase. The authors should mention about these apyrases by citing the following papers:

https://doi.org/10.1094/MPMI.2000.13.10.1053

https://doi.org/10.1007/s10327-006-0279-7

https://doi.org/10.1104/pp.108.117564

We added the appropriate fragment in lines 91-93 and we referred all three above-mentioned references (reference numbers [26], [27], and [28], respectively).

Line 132-133. The following paper should be added as a reference in which the transcriptome in “intensified LecRK-I.9 gene expression” was described.

https://doi.org/10.1080/15592324.2019.1659079

We added the above-mentioned reference (line 136, reference number [35]).

Line 146-150. Tanaka et al. 2014 is better to cite here since the paper summarized entire eATP signaling. https://dx.doi.org/10.3389/fpls.2014.00446

We agree that Tanaka et al. 2014 is better than Nizam et al. 2019, and we change these references. Tanaka et al. 2014 is cited as [41] (line 154) and Nizam et al. 2019 has a new number [50] in the revised version of the manuscript.

Paragraph 189-199. The author should be aware that eATP level also affect on infection of RN rhizobia and AM fungi. It would be nice to describe about it in the paragraph by citing the following paper: https://doi.org/10.1002/9781119053095.ch52

We added the appropriate fragment in lines 202-206 and we cited the above-mentioned reference (reference number [53]).

Lines 446-469 are highly speculative. It is interesting to think of these ideas for signal transduction by uncommon nucleotides, but I think the authors should place more stress on the highly speculative nature of the proposed pathways.

We agree that this is only a supposition/hypothesis and we add a sentence in which it is emphasized that it is our postulate/hypothesis (505-506).

I would appreciate some discussion of the possibility that these uncommon nucleotides signal through their metabolic conversion to ATP. Does this idea make any sense? Does the extracellular ATP receptor mutant dorn1 still respond to these nucleotides?

Currently, it is known only one enzyme that can cleavage only Ap4A giving ATP, so we would like not to discuss this topic in the manuscript. We added in lines 379-380 the information that only Ap4A may be a source of ATP.

It’s surprising that, while they describe extracellular ATP as a signal, they neglected to mention recent research on extracellular pyridine nucleotides, NAD(P), as signals. The authors should cover this topic in the manuscript by citing the relevant papers:

https://doi.org/10.1038/s41467-019-12781-7

https://doi.org/10.1111/j.1365-313X.2008.03687.x

https://doi.org/10.7554/eLife.25474

According to the Reviewer’s requirement, we added section 3. Extracellular Pyridine Nucleotides (lines 226-244) and we referred all three above-mentioned papers (reference numbers [71], [68], and [70], respectively).

Line 463-464. "Many MAPK cascades respond to hormones such as abscisic acid, jasmonic acid, salicylic acid, ethylene, auxins, and brassinosteroids." I think the authors need to cite references for the relevant studies. For example, this paper shows that ABA or JA/coronatine treatment decreases the abundance of phosphorylated MAPKs after flg22 treatment by a transcriptional mechanism:

https://www.pnas.org/content/early/2017/06/23/1702613114

We did not add any comments, but we cited the above-mentioned paper (line 505, reference number [207]).

Is there any place for p4A to fit in the working model in Fig5?

We did not place p4A in Fig. 5 because there is no information about the effect of p4A on phenylpropanoid pathway gene expression.

Line 354-445. Some of the nucleotide effects seem as an intracellular function. The authors carefully describe all statements by specifying whether the effect is as an extracellular signal or an intracellular signal.

Frankly speaking, we do not know what the Reviewer had in mind writing this comment, thus we did nothing at this point.

Line 483-486. The author should cite the following paper in which molecular signal signatures and physical signal signatures are well described and discussed.

https://doi.org/10.1016/j.plantsci.2019.03.004

We added the appropriate fragment in lines 529-533 and we cited the above-mentioned paper (reference number [209]).

According to the reviewers’ suggestions/requirements, we added 11 new references. Besides these 11 references, we add another 11 references. All together is 22 references added to the manuscript. All of them are listed in the References.